# Key Members of the CmPn as Biomarkers Distinguish Histological and Immune Subtypes of Hepatic Cancers

**DOI:** 10.3390/diagnostics13061012

**Published:** 2023-03-07

**Authors:** Johnathan Abou-Fadel, Victoria Reid, Alexander Le, Jacob Croft, Jun Zhang

**Affiliations:** Department of Molecular and Translational Medicine (MTM), Texas Tech University Health Science Center El Paso, El Paso, TX 79905, USA

**Keywords:** liver cancer, liver cancer subtypes, immunological subtypes, classic nuclear progesterone receptors, non-classic membrane progesterone receptors, biomarkers, CCM signaling complex (CSC), tumorigenesis, progesterone, mPRs-PRG-nPRs (CmPn) and CSC-mPRs-PRG (CmP) signaling network

## Abstract

Liver cancer, comprising hepatocellular carcinoma (HCC) and cholangiocarcinoma (CCA), is a leading cause of cancer-related deaths worldwide. The liver is a primary metabolic organ for progesterone (PRG) and PRG exerts its effects through classic nuclear PRG receptors (nPRs) and non-classic membrane PRG receptors (mPRs) or a combination of both. Previous studies have shown that the CCM signaling complex (CSC) couples both nPRs and mPRs to form the CmPn (CSC-mPR-PRG-nPR) signaling network, which is involved in multiple cellular signaling pathways, including tumorigenesis of various cancers. Despite advances in treatment, 5-year survival rates for liver cancer patients remain low, largely due to the chemoresistant nature of HCCs. The lack of sensitive and specific biomarkers for liver cancer diagnosis and prognosis emphasizes the need for identifying new potential biomarkers. We propose the potential use of CmPn members’ expression data as prognostic biomarkers or biomarker signatures for the major types of hepatic cancer, including HCCs and CCAs, as well as rare subtypes such as undifferentiated pleomorphic sarcoma (UPS) and hepatic angiosarcoma (HAS). In this study, we investigated the CmPn network through RNAseq data and immunofluorescence techniques to measure alterations to key cancer pathways during liver tumorigenesis. Our findings reveal significant differential expression of multiple CmPn members, including CCM1, PAQR7, PGRMC1, and nPRs, in both HCCs and CCAs, highlighting the crucial roles of mPRs, nPRs, and CSC signaling during liver tumorigenesis. These key members of the CmPn network may serve as potential biomarkers for the diagnosis and prognosis of liver cancer subtypes, including rare subtypes.

## 1. Introduction

Liver cancer is a significant cause of cancer-related death, ranking third globally, while also being the sixth most commonly diagnosed cancer [1]. The two primary types of liver cancer are hepatocellular carcinoma (HCC) and cholangiocarcinoma (CCA), which, respectively, account for 75% and 12–15% of all cases [2]. The five-year survival rate for liver cancer patients ranges from 5 to 30% [3], with HCC and CCA patients having survival rates of approximately 15% and 5–15%, respectively [4]. Early treatment significantly increases the likelihood of longer recurrence-free periods and better overall survival rates for HCC patients compared to those diagnosed with rarer subtypes of liver cancer, which often have poorer prognoses due to their “chemoresistant” nature [5,6]. Postoperative recurrence of HCC is associated with several major prognostic factors such as tumor size, number of nodules, vascular invasion, tumor encapsulation, blood transfusion, high α-fetoprotein (AFP) levels, and resection margin status [7]. The incidence of liver cancer is higher in males than females worldwide, potentially due to factors such as alcohol consumption, hepatitis B virus (HBV), hepatitis C virus (HCV), or nonalcoholic fatty liver disease (NAFLD) [8]. While most CCA cases occur sporadically, risk factors such as HBV, HCV, infection-associated cirrhosis, diabetes, and obesity have been associated with an increased risk of CCA [9]. Rare subtypes of liver cancer have worse prognoses and fewer successful treatment options due to their low incidence, making it difficult to diagnose them early before metastasis occurs [10,11]. Additionally, the role of progesterone (PRG) signaling cascades is often ignored despite liver cancer being a male-dominant disease. Recent research has explored the potential role of the sex steroid hormone PRG in the progression of hepatocellular carcinoma (HCC) [12,13,14,15]. PRG can interact with classic nuclear PRG receptors (nPRs) or non-classic membrane PRG receptors (mPRs) through classic, non-classic, or mixed responses [16,17,18]. The CSC, which is comprised of KRIT1 (CCM1), MGC4607 (CCM2), and PDCD10 (CCM3), has been observed to interact with mPRs and nPRs to form the CSC-mPR-PRG-nPR (CmPn) signaling network under PRG actions [19,20,21,22]. This network is involved in angiogenesis and tumorigenesis of multiple types of cancers, including liver and breast cancers [16,17,22,23,24,25,26,27,28]. 

Among the mPRs, mPRα (PAQR5), mPRβ (PAQR6), mPRγ (PAQR7), mPRδ (PAQR8), and mPRε (PAQR9) have been investigated for their involvement in angiogenesis and tumorigenesis [29,30]. Therefore, the alterations to key members of the CmPn signaling network can be further evaluated as potential biomarkers for not only HCCs and CCAs but also for rarer subtypes of hepatic cancers. Progesterone receptor membrane component 1 (PGRMC1), a non-classical membrane progesterone receptor, has been associated with the progression of HCC from tumor size G2 to G3 and has been explored as a possible prognostic biomarker for HCCs [31,32]. High CCM3 gene expression has also been linked to poor prognosis due to its promotion of cell proliferation and metastasis in HCC [33]. In addition, altered expression of all three members of the CSC, along with PAQR7, has been observed during the early stages of tumorigenesis in liver cancer [26]. Currently, there is a lack of non-invasive diagnostic and prognostic measures for hepatic cancer that can differentiate between the various subtypes with distinct histology and morphology. Alpha-fetoprotein (AFP) is the current gold standard biomarker for diagnosing liver cancer [34], but it has variable sensitivity and specificity and is often used as a complementary biomarker [35]. The lack of sensitive and specific biomarkers for the prognosis and diagnosis of liver cancer underscores the urgent need for new potential biomarkers that can be utilized for earlier diagnosis and non-invasive prognostic methods to distinguish between liver cancer subtypes.

In this study, we examined the mRNA expression of essential CmPn players, including the CSC (CCM1-3), mPRs (PAQR5-9, PGRMC1/2), and nPRs (PGR1/2), along with the current liver biomarker AFP, across various liver tissue and cancer types, along with patient clinical information. In addition, we utilized immunofluorescence (IF) techniques to analyze protein expression of essential CmPn members, such as CCM1/3, PAQR7/8, PGRMC1, and nPRs. Our results demonstrate that several key members of the CmPn network have the potential to be utilized as biomarkers for diagnosing and prognosing different liver cancer subtypes at both the transcriptional and translational levels, as revealed through our comparative bioinformatics and IF approaches.

## 2. Materials and Methods

### 2.1. Bioinformatics Analysis

Clinical profiling for hepatic cancers utilizing the NCI-GDC data portal. Utilizing the Genomic Data Commons (GDC) data portal from the National Cancer Institute (NCI), we assessed all available sociological/clinical/diagnostic data for patients diagnosed with liver (Hepatocellular carcinoma, HCC) or intrahepatic bile duct cancers (Cholangiocellular carcinoma, CCA) [36,37,38,39,40,41]. We performed preliminary general clinical analysis for HCC and CCA patients, which utilized 14 publicly available databases (Appendix A) and repeated our clinical analysis for CmPn network associated clinical observations utilizing only 2 databases (Appendix A) containing patient samples with differential expression data for any of the key CmPn players.

Differential expression profiling of key CmPn members, along with *AFP*, using microarray expression data. Expression analysis was performed as previously described [16,17,23,26]. Briefly, differential expression profiling was preliminarily performed utilizing the TCGA-TARGET-GTEx database, which is a unique database that contains two types of ‘normal’ tissues’: (1). “solid tissue normal” which are taken from normal tissue, near the tumor site, and (2). ‘normal healthy tissue’ from individuals without cancer. These data originate from the UCSC RNAseq compendium, where TCGA, TARGET, and GTEx samples are re-analyzed (re-aligned to human hg38 reference genome and expressions are processed using RSEM and Kallisto methods) by the same RNA-seq pipeline, thereby eliminating batch effects. Additionally, the TCGA-PANCAN database was also used to validate results obtained using the TCGA-TARGET-GTEx database. Available sociological, pathological, and follow-up clinical data [42], for both HCC and CCA patients, were extracted from the TCGA-LIHC and TCGA-PANCAN databases and used during expression profiling to investigate differentially expressed genes among race, gender, family cancer history, vascular invasion, stemness scores across histological types, new tumor events (type and site), vital status, and residual tumor status [42].

Additionally, RNAseq data from the TCGA-TARGET-GTEx, as well as the TCGA-PANCAN databases, were used to investigate differential expression of key CmPn members, along with *AFP*, for both HCCs and CCAs, based on histological sample type (normal healthy tissue vs. primary tumor samples). We then repeated expression profiling for each cancer type based on immune subtypes, in a pair-wise fashion. Finally, we evaluated expression profiling of key CmPn members, along with *AFP*, between HCCs and CCAs, to evaluate overlaps and differences in expression between these two cancer subtypes. Significant differential expression was performed using Xenabrowser with a cutoff *p*-value <0.05. Gene ontology, KEGG, and reactome pathways enrichment analysis was performed from the RNAseq data using Enrichr. Significance was evaluated using a cutoff *p*-value <0.05 and FDR cutoff value <0.1.

### 2.2. Immunofluorescence (IF)

*Deparaffinization of paraffin-embedded tissue sections.* IF processing of slides was performed as previously described [16,17,23,26]. Briefly, liver tissue microarray slides were purchased from US Biomax (Appendix A) and were baked at 60 °C for 2 h. Once the slides cooled, the sections underwent 3 xylene washes at 5 min each, followed by sequential 3 min washes of 100, 95, 90, 80, and 70% ethanol and soaked in water before undergoing antigen retrieval.

*Antigen Retrieval.* Tissue sections were washed in 1X PBS 3 times, 3 min per wash, before being permeabilized in PBS containing 0.2% Triton X-100 (wash buffer) for 10 min. Tissue slides were then submerged in 10mM sodium citrate buffer (Na_3_C_6_H_5_O_7_, pH 6.0) containing 0.01% Triton X-100. Tissues were maintained at 95–98 °C for 30 min in citrate buffer and then set aside to cool at room temperature (RT).

*Blocking and Antibody Incubation.* Following antigen retrieval, slides were washed 3 times in wash buffer at 3 min per wash at RT. The tissues were blocked using Pierce Fast Blocker Buffer (Fisher) for 90 min at room temperature to optimize the blocking process. A hydrophobic pen was used to draw a barrier around the tissue sections to conserve antibodies. All primary antibodies were diluted in wash buffer and Pierce Fast Blocking Buffer (1:1). An initial 500 uL was added to the tissues and left to incubate for 2 h at RT in the dark. Primary antibody was removed, and tissues were washed 3 times in wash buffer at 3 min per wash at RT. An amount of 400 uL of secondary antibody, which was diluted in wash buffer and Pierce Fast Blocking Buffer (1:1), was added to the slide and left to incubate for 2 h at RT. All antibodies used are listed in Appendix A. Tissues were washed 3 times in wash buffer at 3 min per wash at RT before continuing to the mounting/sealing step.

*Nuclear Staining and Mounting/Sealing.* Tissue sections were stained with DAPI by adding two drops of mounting media containing DAPI directly to the sections. Tissues were left to rest overnight at 4 °C in the dark to allow efficient staining of DAPI. Slides were sealed with nail polish the next day and allowed time to dry before imaging.

*Imaging and Quantification.* IF imaging/quantification of microarray slides was performed as previously described [16,17,23,26]. Briefly, imaging was performed utilizing a Nikon Eclipse Ti confocal microscope using 10×, 20×, and 60× objective lenses. Nikon Elements Analysis software, equipped on the Nikon microscope, was used to automatically quantify protein abundance in the tissues. Bias was avoided by maintaining thresholding values across all images and to dismiss low and high outliers. IF images were quantified for CCM1 and PGRMC1 using 488 nm wavelength channel, while images for CCM3, PAQR7/8, nPRs, and AFP were quantified using the 555 nm channel.

### 2.3. Prognostic Outcomes for Key CmPn Members Using HCC Patient Samples

Construction of Kaplan–Meier (KM) survival curves to determine prognostic effects. Publicly available microarray data (22,277 probes) from 364 liver cancer patients (only available for HCCs due to small sample size for CCAs) were analyzed using KMplotter [43] to integrate gene expression and clinical data simultaneously. To ensure patients in the database reflected cohorts seen in the everyday clinical practice, we filtered the patient data by only selecting cohort data similar to SEER published prevalence [43]. Additionally, publicly available microarray data from 374 liver cancer patients (again, only available for HCCs due to small sample size for CCAs) were also assessed from TCGA to integrate gene expression and clinical data simultaneously [42] to confirm the initial analysis performed using KMplotter. Log-rank *p*-values, hazard ratios, and 95% confidence intervals were calculated by the software [43].

### 2.4. Statistical Analysis

For IF analysis, one-way analysis of variance (ANOVA) or two-way ANOVA (for co-stained sections) was used to detect differences in the mean abundance of proteins among the various liver tissue types with Holm–Sidak’s multiple comparisons correction. Welch’s *t*-test was used to detect the differences in the mean values among two comparing groups. For bioinformatics analysis, Welch’s *t*-test was also used to detect the differences in the mean values among two comparing groups, while one-way ANOVA was used to detect the differences in the mean values among more than two groups. All IF graphs/plots/charts were constructed and produced using GraphPad Prism 9.3.1, while bioinformatics graphs/plots/charts were produced using either the Xena, KMplotter, or NCI-GDC built-in analysis software.

## 3. Results

### 3.1. Clinical Profiling of Hepatic Cancers with Altered CmPn Expression

Given our recent findings of the potential role of the CmPn signaling network in tumorigenesis of multiple types of cancers [16,17,23,26], we began our investigation by assessing general clinical information for hepatic cancers utilizing the Genomic Data Commons (GDC) data portal from the National Cancer Institute (NCI). We then compared these results to filtered data, only analyzing patient samples with differential expression for any key CmPn players (Figure 1, Appendix A). By filtering the NCI-GDC data to look only at patients with differential CmPn expression, we observed that approximately 80% of hepatic cancer patients with altered expression of key CmPn members had a life expectancy of ~9 years, compared to ~10 years for hepatic cancer patients without modified CmPn expression (Figure 1(A-1),(A-5), left panels). It was also noted that patients with CmPn alterations were diagnosed later in life ~61 years old, compared to ~58 years for hepatic cancer patients without modified CmPn expression (Figure 1(A-1),(A-5), right panels). Both year of diagnosis (Figure 1(A-2),(A-6)) and year of death analyses (Figure 1(A-4),(A-8)) were similar, indicating that regardless of differential expression of key CmPn members, there has been an exponential increase in both categories for hepatic cancer patients. Interestingly, when evaluating tissue of origin (Figure 1(A-3),(A-7)), there was a noticeable increase in the number of cases originating in the intrahepatic bile ducts for patients with differential expression of CmPn members (~12%) compared to patients without CmPn alterations (~3%).

We additionally performed socioeconomic status (SES) and diagnostic comparative profiling utilizing clinical data for patient samples without (Figure 1(B1–B5)) or with differential expression for key CmPn players (Figure 1(B6–B9)). Our analysis revealed that the majority of patients diagnosed with hepatic cancers were not Hispanic/Latino, irrespective of CmPn members’ expression (Figure 1(B-1),(B-6), left panels). The percentage of patients among the four primary racial categories between our two groups were relatively similar, although with a higher number of Asian patients (~40%) with altered CmPn expression data (Figure 1(B-6), right panel) compared to patients without CmPn alterations (~14%; Figure 1(B-1), right panel). Gender stratification of patients illustrated that hepatic cancer is dominant in males (Figure 1(B-2),(B-7)), regardless of altered expression of key CmPn members. Interestingly, an 11% increase was observed in patients with prior malignancy with differential expression of CmPn genes (Figure 1(B-8)) compared to those without (Figure 1(B-3)). Additionally, when we examined primary diagnosis status, we saw that nearly half of cases were identified as either CCA or HCC, while <1% were recognized as either combined hepatocholangiocarcinoma (cHCC-CCA) or clear cell adenocarcinoma (CCC) (Figure 1(B-4)) for patients without CmPn alterations. Interestingly, when we repeated the analysis for patients with altered CmPn expression, we observed a switch in the primary diagnosis classifications in which now ~80% were diagnosed with HCC and only ~10% with CCA (Figure 1(B-9)), while cHCC-CCA and CCC remained the least diagnosed subtypes. Finally, we conducted an assessment on general tumor classification data for patients without CmPn expression (not enough data for altered CmPn expression analysis) and observed more hepatic cancer cases that began as primary cancers rather than a result of metastasis from another organ (Figure 1(B-5)). Overall, these findings suggest that altered CmPn expression may have an impact on certain clinical aspects of hepatic cancers.

### 3.2. Altered Expression of CmPn Genes across Clinical Tumors in Hepatic Cancers Suggest Their Involvement in Liver Cancer Tumorigenesis

Based on our preliminary clinical analysis of altered SES and diagnostic profiling between patients with and without differential expression of key CmPn players, we next evaluated expression of CmPn players in liver tissues using publicly available RNAseq data from The Cancer Genome Atlas (TCGA). To evaluate differential expression of key CmPn players and their potential role as prognostic biomarkers, we also profiled AFP, an established liver cancer biomarker, to compare with our CmPn profiling analysis. For our first analysis, we utilized two databases in TCGA containing two categories of ‘normal’ tissues: “solid tissue normal” which are normal tissue sections taken adjacent to primary tumor tissue and “normal healthy tissue” taken from healthy individuals with no diagnosis of cancer. We observed significant differential expression patterns, in both HCCs and CCAs, for almost all CmPn players (except *PGRMC2* in CCAs), along with *AFP,* in our preliminary analysis (Figure 2(A-1),(A-2)). Next, we repeated expression profiling using a separate database in TCGA, based on tissue type (normal healthy tissue unavailable in this database), confirming several of our previous observations. We confirmed significant differential expression patterns for *CCM2*, *PAQR6/8*, *PGRMC1/2*, and *nPRs*, along with *AFP,* for HCCs (Figure 2(B-1)), while analysis of CCAs (recurrent tumor data unavailable) confirmed significant differential expression patterns for almost all CmPn players (except *CCM1/PAQR7*), along with *AFP* (Figure 2(B-2)). 

To further validate the role of the CmPn network during liver cancer tumorigenesis, we analyzed differentially expressed genes (DEGs) in a pair-wise fashion, comparing normal healthy liver tissue to HCC primary tumor tissues, using publicly available RNAseq data. After analyzing DEGs between 110 normal healthy liver tissues and 369 HCC primary tumor tissues (Appendix A), we observed down regulation of *CCM1/2*, *PAQR6*, and *nPRs* in HCC primary tumor tissues, while all other CmPn players were up regulated, along with *AFP* (Figure 2(C-1)). We then utilized all DEGs to perform pathway enrichment analysis to display the top 15 affected biological processes, molecular functions, and KEGG pathways between HCC primary tumor tissues and normal healthy liver tissues (Figure 2(C-2) and Appendix A). Our results illustrated that up-regulated DEGs in HCC primary tumors significantly affected cell cycle regulation, protein binding, ATP-dependent activity, histone phosphorylation and cancer signaling pathways including DNA replication, spliceosomes/lysosomes, pancreatic cancer, and hormone signaling pathways including oocyte meiosis and progesterone-mediated oocyte maturation (Figure 2(C-2), left panels and Appendix A). Down-regulated DEGs significantly affected actin filament-based processes, protein digestion and absorption (also seen in a recent analysis of HCC tumors using four separate independent databases) [44], and pancreatic secretion (Figure 2(C-2), right panels and Appendix A). Interestingly, dysfunction of pancreatic secretion has been reported as a metabolic change resulting from liver cancer [45,46]. Several important cancer signaling pathways were also observed to be affected (with significant *p*-values, but FDR > 0.1) including P53 signaling, viral carcinogenesis, colorectal cancer, base excision repair, and the Fanconi anemia pathway (Figure 2(C-2), lower left panel and Appendix A). 

We repeated DEG analysis this time comparing 110 normal healthy liver tissues to 36 CCA primary tumor tissues, which demonstrated more overall down-regulated CmPn genes in CCA primary tumors, compared to our HCC analysis (Figure 2(D-1) and Appendix A). We observed down regulation of *CCM1/2* (down in HCC), *PAQR9* (up in HCC)*, PGRMC1/2* (up in HCC), *nPRs* (*down in HCC*), and *AFP* (up in HCC) in CCA primary tumors. Interestingly, *CCM3* and *PAQR5/7/8* were up regulated in CCAs (Figure 2(D-1) and Appendix A), which were also up regulated in our HCC analysis. Utilizing DEGs from CCAs, we performed pathway enrichment analysis and observed that up-regulated DEGs in CCA primary tumors were significantly enriched in mainly regulation of cell communication, cell population proliferation, cell differentiation, cell junction assembly, and regulation of Wnt signaling (Figure 2(D-2), left panels and Appendix A). KEGG pathway analysis also revealed that several important cancer signaling pathways were observed to be up regulated (with significant *p*-values, but FDR > 0.1) including PI3K-Akt, Notch, ECM–receptor interactions, regulation of actin cytoskeleton, tight junction, focal adhesion, and adherens junction signaling pathways (Figure 2(D-2), lower left panel and Appendix A), which have been previously observed in breast cancer studies with disruption of the CmPn network [16,17,23]. Down-regulated DEGs mainly affected cholesterol/sterol/lipid homeostasis and transporting processes, hydrolase/transferase activity, cholesterol metabolism, regulation of blood coagulation, several cancer signaling cascades including complement and coagulation cascades, PPAR pathway, bile secretion, peroxisomes, and ABC transporters, as well as metabolic signaling pathways including fat digestion/absorption (Figure 2(D-2), right panels and Appendix A). 

Lastly, to validate diagnostic potential of key CmPn players, we analyzed DEGs comparing both primary liver tumor subtypes. After profiling DEGs between 438 HCC tumor samples and 45 CCA tumor samples, we observed down regulation of *CCM2*, *PAQR9*, and *PGRMC1/2*, along with *AFP,* in CCA tumor tissues, while *CCM1*, *nPRs*, and *PAQRs5-8* were up regulated in CCA tissues (Figure 2(E-1) and Appendix A). Pathway enrichment analysis revealed up regulation of mainly cellular localization, cellular component biogenesis, transmembrane transport processes, transmembrane signaling receptors, frizzled binding, G protein-coupled receptors, MAPKKK cascades, cell–cell adhesion, and proton transmembrane transport functions in CCAs (Figure 2(E-2), left panels and Appendix A). KEGG pathway analysis of CCA tumor tissues revealed several up-regulated pathways (with significant *p*-values, but FDR > 0.1) important in cancer signaling including Wnt, AMPK, PI3K-Akt, breast cancer, prostate cancer, ECM–receptor interactions, and proteoglycans in cancer signaling pathways (Figure 2(E-2), left bottom panel and Appendix A). Down-regulated DEGs significantly impacted exocytosis, sterol homeostasis, lipid/sterol transport, cholesterol/sterol metabolism, hemostasis, blood coagulation, wound healing processes, lipase/endopeptidase activity, regulation of proteolysis, sterol binding/transport, and lipid localization (Figure 2(E-2), right panels and Appendix A). Similar to the analysis performed with CCA tumors and normal healthy tissue, down-regulated signaling pathways in CCAs when compared to HCCs also included cholesterol metabolism, several cancer signaling cascades including chemical carcinogenesis, complement and coagulation cascades, PPAR pathway, bile secretion/synthesis, peroxisomes, and ABC transporters, as well as metabolic signaling pathways including fat digestion/absorption (Figure 2(E-2), right bottom panel and Appendix A). In sum, these data suggest that CmPn genes may have a potential diagnostic and prognostic role in liver cancer.

### 3.3. Differential Expression Patterns of Key CmPn Genes across Various Histological Categories of Hepatic Cancers and Their Associated Prognostic Effects

Utilizing TCGA databases, we compiled expression profiles for hepatic cancers by filtering histological types comparing HCC, the three major types of CCA (intrahepatic, distal, and hilar/perihilar), and cHCC-CCAs. After comparing all subtypes of CCA to HCC and cHCC-CCA clinical tumors, we observed significant differential expression of almost all CmPn members, as well as *AFP*, except for *CCM3* (Figure 3A). Recently, there has been mounting evidence that tumor cells can exhibit stem cell-like properties, allowing tumor cells the capacity of self-renewal, which is responsible for the long-term maintenance of tumors including therapeutic resistance, tumor dormancy, and metastasis [47,48,49]. Given the importance of this, we next assessed stemness scores between HCCs, the three subtypes of CCAs, and cHCC-CCAs. Utilizing clinical RNAseq data, we obtained significant differences between stemness scores among the three major subtypes of CCAs, HCCs, and cHCC-CCA clinical tumors (Figure 3B), suggesting that all subtypes analyzed have significant differences in terms of self-renewal and repopulation capacities. Again, these results reaffirm that altered CmPn gene expression may serve as potential biomarkers for hepatic cancers and that targeting the CmPn network may be a promising therapeutic strategy for liver cancer.

### 3.4. Differential Expression Patterns of key CmPn Genes across Immune Subtype Classifications among Hepatic Cancers and Their Associated Prognostic Effects

Given the importance of tumor microenvironments for both tumorigenesis and immunogenicity [50], we analyzed differential expression of CmPn network genes, along with *AFP*, among immune subtypes for both HCCs and CCAs. Clinical data available for HCC and CCA tumors included immune subtype classifications for wound healing (C1), IFN-gamma dominant (C2), inflammatory (C3), and lymphocyte depleted (C4) subtypes. Overall expression profiling of 355 HCC tumors, across all immune subtypes, revealed significant differential expression patterns for almost all CmPn players, along with *AFP*, excluding *CCM2* and *PAQR7/9* (Figure 3(C-1)). When we repeated our analysis for overall expression profiling of 35 CCA tumors, across all immune subtypes, we only obtained significant differential expression patterns for *PAQR6*, which may be attributable to a small sample size of available clinical data for CCAs, in TCGA, regarding immune subtype classifications (Figure 3(C-2)). These results not only illustrate the involvement of the CmPn network in the tumor microenvironment, but also demonstrate the potential use of expression profiling among multiple members of the CmPn signaling network for HCCs, and *PAQR6* for CCAs, as potential prognostic biomarkers across various immune subtypes in hepatic cancers. 

#### 3.4.1. Differential Expression Patterns of mPR and CCM Genes between Wound Healing (C1) and IFN-γ Dominant (C2) Immune Subtype Classifications 

To further investigate the significance of immune subtypes among HCCs and CCAs, we performed pair-wise comparisons between various immune subtypes, given their importance in influencing the tumor microenvironment. First, we compared wound healing (C1), characterized by elevated expression of angiogenic genes, a high proliferation rate, and a Th2 cell bias to the adaptive immune infiltrate, with IFN-γ dominant (C2) immune subtype, which has the highest M1/M2 macrophage polarization, a strong CD8 signal, high proliferation rate, and high T-cell receptor diversity [51]. Analysis of HCCs demonstrated mainly up regulation in C1 subtype for members of the CSC and mPRs (thereby down regulated in C2 subtype), with significance obtained for *CCM1* and *PAQR6* (Appendix A, left upper panel and Appendix A). Interestingly, pathways functional enrichment analysis (Appendix A) revealed mainly down-regulated pathways for C1 subtype (thereby up regulated in C2 subtype) including chemokine receptor binding, MHC protein binding, and antigen processing and presentation (Appendix A, lower left panel and right panels). 

Interestingly, when we repeated this analysis for CCAs, we did not observe any significant differential expression for key CmPn members (Appendix A and Appendix A), most likely due to a limited sample size. 

#### 3.4.2. Differential Expression Patterns of mPR and CCM Genes between Wound Healing (C1) and Inflammatory (C3) Immune Subtype Classifications 

We next compared C1 subtype with the inflammatory immune subtype (C3), characterized by elevated Th1/Th17 genes, low to moderate tumor cell proliferation, lower levels of aneuploidy, and lower overall somatic copy number alterations than the other immune subtypes [51]. Analysis of HCCs demonstrated significant up regulation in C1 subtype (thereby down regulated in C3 subtype) for *CCM3* and *PAQR5/6*, along with *AFP* (Appendix A, left upper panel and Appendix A). We only observed significant down regulation (thereby up regulated in C3 subtype) for *PGRMC1/2* (Appendix A, left upper panel and Appendix A). Interestingly, pathways functional enrichment analysis (Appendix A) revealed mainly up regulation for C1 subtype (thereby down regulated in C3 subtype) for p53 signaling, Fanconi anemia pathway, and progesterone-mediated oocyte maturation (Appendix A, lower left and right panels). Down-regulated pathways for C1 subtype (thereby up regulated in C3 subtype) mainly included fatty acid metabolic processes, oxidoreductase activity, and complement/coagulation cascades (Appendix A, lower left and right panels). 

When we repeated this analysis for CCAs, with a much smaller sample size, we observed significant down regulation in C1 subtype (thereby up regulated in C3 subtype) for *PAQR6* (Appendix A, left upper panel and Appendix A), which interestingly was up regulated for HCCs in the same comparison (Appendix A, left upper panel). Pathways functional enrichment analysis (Appendix A) in CCAs revealed significant up-regulated pathways in C1 subtype (thereby down regulated in C3 subtype) for PPAR, IL-17, and HIF-1 signaling pathways (Appendix A, lower left panel and right panels). Interestingly, there were very few significant down-regulated pathways observed in CCAs which included oxidoreductase activity, bile secretion, and Rap1 signaling pathways (Appendix A, lower left panel and right panels). 

#### 3.4.3. Differential Expression Patterns of mPR and CCM Genes between Wound Healing (C1) and Lymphocyte Depleted (C4) Immune Subtype Classifications 

We then compared C1 subtype with lymphocyte depleted immune subtype (C4), characterized by a more prominent macrophage signature, with Th1 suppressed and a high M2 macrophage response [51]. Analysis of HCCs demonstrated significant up regulation in C1 subtype (thereby down regulated in C4 subtype) for *PAQR5/6/8*, along with *AFP* (Appendix A, left upper panel and Appendix A). Significant down regulation in C1 subtype (thereby up regulated in C4 subtype) was observed for *PGRMC1/2*, *PAQR9*, and *CCM2* (Appendix A, left upper panel and Appendix A). Interestingly, pathways functional enrichment analysis (Appendix A) revealed mainly up regulation for C1 subtype (thereby down regulated in C4 subtype) for Wnt signaling, ECM proteoglycans in cancer pathways, and Hippo signaling pathways (Appendix A, lower left and right panels). Down-regulated pathways in C1 subtype (thereby up regulated in C4 subtype) were more prominent including fatty acid/acyl-CoA metabolic and ligase processes, cholesterol metabolism, and PPAR signaling pathways (Appendix A, lower left panel and right panels). 

When we repeated this analysis for CCAs, with a much smaller sample size, we observed significant down regulation in C1 subtype (thereby up regulated in C4 subtype) for *PAQR6* (Appendix A, left upper panel and Appendix A), which was up regulated for HCCs in the same comparison (Appendix A, left upper panel). Pathways functional enrichment analysis (Appendix A) in CCAs revealed significant up-regulated pathways in C1 subtype (thereby down regulated in C4 subtype) for cytokine activity, TNF, IL-17, and PIK3-Akt signaling pathways (Appendix A, lower left panel and right panels). Significantly down-regulated pathways in C1 subtype (thereby up regulated in C4 subtype) included mitochondrial translational pathways, non-alcoholic fatty liver disease signaling, and shared signaling cascades with other diseases including Parkinson’s/Alzheimer’s/Huntington’s disease (Appendix A, lower left panel and right panels). 

#### 3.4.4. Differential Expression Patterns of mPR, nPRs, and CCM Genes between IFN-γ Dominant (C2) and Inflammatory (C3) Immune Subtype Classifications 

We next compared C2 with C3 immune subtypes. Analysis of HCCs demonstrated significant up regulation in C2 subtype (thereby down regulated in C3 subtype) for *CCM3* and *PAQR6/8*, along with *AFP* (Appendix A, left upper panel and Appendix A). Significant down regulation in C2 subtype (thereby up regulated in C3 subtype) was observed for *PGRMC1/2*, *nPRs*, and *CCM1* (Appendix A, left upper panel and Appendix A). Pathways functional enrichment analysis (Appendix A) revealed significant up regulation for C2 subtype (thereby down regulated in C3 subtype) for cell cycle phase transition, chemokine activity, and TH1/TH2 cell differentiation pathways (Appendix A, lower left panel and right panels). Down-regulated pathways in C2 subtype (thereby up regulated in C3 subtype) included Notch signaling, breast/endometrial/prostate cancer signaling pathways, mTOR, and FoxO signaling pathways (Appendix A, lower left panel and right panels). 

When we repeated this analysis for CCAs, with a much smaller sample size, we observed significant down regulation in C2 subtype (thereby up regulated in C3 subtype) for *PAQR5/6* and *PGRMC1,* along with significant up regulation of *CCM2* (Appendix A, left upper panel and Appendix A). Pathways functional enrichment analysis (Appendix A) in CCAs revealed significant up-regulated pathways in C2 subtype (thereby down regulated in C3 subtype) for T-cell proliferation/activation, cytokine/chemokine signaling, and lymphocyte differentiation pathways (Appendix A, lower left panel and right panels). Significantly down-regulated pathways in C2 subtype (thereby up regulated in C3 subtype) included cell adhesion molecules (CAMs), PI3K-Akt signaling, and carbohydrate digestion/absorption pathways (Appendix A, lower left panel and right panels). 

#### 3.4.5. Differential Expression Patterns of mPR and CCM Genes between IFN-γ Dominant (C2) and Lymphocyte Depleted (C4) Immune Subtype Classifications 

We next compared C2 with C4 immune subtypes. Analysis of HCCs demonstrated significant up regulation in C2 subtype (thereby down regulated in C4 subtype) for *PAQR5/8* along with *AFP* (Appendix A, left upper panel and Appendix A). Significant down regulation in C2 subtype (thereby up regulated in C4 subtype) was observed for *PGRMC1/2* and *CCM1* (Appendix A, left upper panel and Appendix A). Pathways functional enrichment analysis (Appendix A) revealed significant up regulation for C2 subtype (thereby down regulated in C4 subtype) for neutrophil activation/degranulation, hemostasis, and CAMs pathways (Appendix A, lower left panel and right panels). Down-regulated pathways in C2 subtype (thereby up regulated in C4 subtype) included fatty acid processes, bile acid biosynthesis, and PPAR signaling (Appendix A, lower left panel and right panels). 

When we repeated this analysis for CCAs, with a much smaller sample size, we observed significant down regulation in C2 subtype (therefore up regulated in C4 subtype) for *PAQR6* and *PGRMC1* (Appendix A, left upper panel and Appendix A). Pathways functional enrichment analysis (Appendix A) in CCAs revealed significant up-regulated pathways in C2 subtype (thereby down regulated in C4 subtype) for B-cell activation and lymphocyte differentiation pathways (Appendix A, lower left panel and right panels). Significantly down-regulated pathways in C2 subtype (thereby up regulated in C4 subtype) included VEGFR binding/functions, cadherin binding, and SIRT1 negative regulation of rRNA pathways (Appendix A, lower left panel and right panels). 

#### 3.4.6. Differential Expression Patterns of mPR, nPRs, and CCM Genes between Inflammatory (C3) and Lymphocyte Depleted (C4) Immune Subtype Classifications 

Finally, we performed pair-wise comparisons between C3 and C4 immune subtypes. Analysis of HCCs demonstrated significant up regulation in C3 subtype (thereby down regulated in C4 subtype) for *PAQR5/8*, along with *nPRs* (Appendix A, left upper panel and Appendix A). Significant down regulation in C3 subtype (thereby up regulated in C4 subtype) was observed for *CCM2/3* (Appendix A, left upper panel and Appendix A). Pathways functional enrichment analysis (Appendix A) revealed significant up regulation for C3 subtype (thereby down regulated in C4 subtype) for angiogenesis, ERK1/2 and MAPK cascades, as well as PI3K-Akt signaling (Appendix A, lower left panel and right panels). Down-regulated pathways in C3 subtype (thereby up regulated in C4 subtype) included G2/M phase transition, Fanconi anemia pathway, and ubiquitin protein activities (Appendix A, lower left panel and right panels). 

When we repeated this analysis for CCAs, with a much smaller sample size, we only observed significant down regulation in C3 subtype (thereby up regulated in C4 subtype) for *CCM3* (Appendix A, left upper panel and Appendix A). Pathways functional enrichment analysis (Appendix A) in CCAs revealed significant up-regulated pathways in C3 subtype (thereby down regulated in C4 subtype) for Notch signaling, blood coagulation, and semaphorin interactions (Appendix A, lower left panel and right panels). Significantly down-regulated pathways in C3 subtype (thereby up regulated in C4 subtype) included non-alcoholic fatty liver disease, glycolysis/gluconeogenesis, and TCA cycle pathways (Appendix A, lower left panel and right panels). Overall, despite a limited sample size of CCAs that leads to fewer significant findings than that for HCCs, these findings suggest that immune subtypes have a significant impact on the expression of genes in the CmPn network and their potential role in the tumor microenvironment.

### 3.5. Expression Profiling of Key CmPn Players across HCC Tumors Integrating SES, Follow-Up and Clinical Survival Data

We repeated expression-profiling patterns for key CmPn players, along with *AFP*, in liver tumor tissues using patient clinical data including race, family cancer history, and vascular invasion, as well as follow-up data regarding new tumor events/sites and vital status. We first evaluated expression profiles based on the four major race categories available. Interestingly, for patients diagnosed with HCCs, significant differential expression of *CCM1/2, PAQR7, PGRMC1/2*, and *nPRs*, along with *AFP*, were observed across the four major races (Figure 3D). Interestingly, when we repeated our analysis for CCAs, we observed no significance among CmPn genes (Appendix A). Next, we performed expression profiling based on family cancer history and observed significant differential expression patterns for *PAQR6, PGRMC1/2*, and *nPRs* among HCC patients (Figure 3E). We then explored expression patterns among genders and observed significant differential expression for *PAQR6* and *nPRs* in HCC patients (Appendix A), with no significance for CmPn genes, except *AFP*, in CCA patients (Appendix A). We next evaluated expression profiling based on vascular invasion and tumor recurrence for HCC clinical tumors. The three main categories for vascular invasion (defined as malignant cells lining the vascular cavities of endothelial cells or portal/hepatic veins) [52] included macro (gross tissue evaluation), micro (histopathological examination of tumor and surrounding hepatic tissue), or none. Significant differential expression patterns for *PAQR5* and *PGRMC1* were observed in HCC patients (Figure 3F), demonstrating the importance of mPRs in influencing vascular invasion. 

We next evaluated differential expression among HCC patients with new tumor events including locoregional recurrence (growth of cancer cells at the same site as the primary tumor), intrahepatic recurrence (recurrence of cancer in the liver after hepatic resection), extrahepatic recurrence (recurrence of the cancer in the lungs, lymph nodes, etc.), new primary tumor, or no new tumor events. Interestingly, we observed significant differential expression in HCC patients for *CCM2*, *PGRMC1*, and *nPRs*, along with *AFP,* demonstrating the importance of the CmPn network in influencing tumor recurrence (Figure 3G). Next, we profiled expression patterns based on new tumor event site and vital status for HCC and CCA patients. Anatomical tumor recurrence between the lungs, liver, lymph nodes, bones, and brain revealed significant differential expression for *CCM1*, *PAQR9*, and *PGRMC1*, along with *AFP,* in HCC patients (Figure 3H), while no significance was observed for CCA patients (Appendix A). Interestingly, we also evaluated residual tumor classification, which evaluates tumor status, and post treatment/resection, for CCA and HCC patients with either micro (R1), macro (R2), or no residual tumors (R0). We observed no significant expression differences for HCC patients (Appendix A), but observed significant expression differences in *CCM2/3*, along with *AFP*, for CCA patients (Appendix A). Finally, for HCC patients, we observed significant differential expression for *PAQR6/7* and *nPRs* based on vital status (Figure 3I), but observed no significant expression differences for CCA patients (Appendix A). Together, these results validate the involvement of the CmPn signaling network in liver cancer tumorigenesis, and furthermore demonstrate their involvement in tumor recurrence mechanisms. In sum, our clinical data support the future use of CmPn members’ expression data, confirmed with *AFP*, as potential prognostic biomarkers for distinguishing between primary subtypes of liver cancer and evaluating tumor recurrence in patients. Given these results, we wanted to further our investigation by assessing whether altered expression of these genes was also observed at the translational level using immunofluorescent techniques. In sum, these integrated analyses with clinical data support the future use of CmPn members’ expression data, along with AFP, as potential prognostic biomarkers for distinguishing between primary subtypes of liver cancer and evaluating tumor recurrence in patients.

### 3.6. Protein Expression Profiling of Key CmPn Members among Normal and Liver Cancer Tissues 

To further examine the involvement of the CmPn network in liver tumorigenesis, we next examined differential expression patterns of key CmPn proteins across seven liver cancer subtypes and normal liver tissues (Appendix A). Utilizing liver tissue microarrays (Appendix A and Appendix A), we analyzed protein expression levels of CCM1/3, PAQR7/8, PGRMC1, and nPRs, along with AFP, using immunofluorescence (IF) imaging (Appendix A). We observed significantly decreased protein expression for CCM1 (Figure 4(A-1),(A-2)), CCM3 (Figure 4(B-1),(B-2)), PAQR8 (Figure 4(C-1),(C-2)), PGRMC1 (Figure 4(D-1),(D-2)), and nPRs (Appendix A) in all liver cancer subtypes, compared to normal healthy liver tissues. Interestingly, PAQR7 displayed similar decreased trends among liver cancer subtypes, with the exception of HCC, which had increased expression compared to normal liver cancer tissues (Figure 4(E-1),(E-2)). When comparing protein expression of key CmPn players within liver cancer subtypes, PAQR7, PGRMC1, and nPRs proteins were significantly differentially expressed only when comparing HCC to all other subtypes (Figure 4D,E and Appendix A), which was also observed in our preliminary TCGA analysis between HCCs and CCAs (Figure 3A). Interestingly, for CCM1/3 and PAQR8, we observed more significant differential expression patterns within all cancer subtypes imaged, with undifferentiated pleomorphic sarcoma (UPS) displaying the lowest levels for CCM3 (Figure 4B) and PAQR8 expression (Figure 4C), while hepatic angiosarcoma (HAS) displayed the weakest expression of CCM1 (Figure 4A), which was also true for PGRMC1 (Figure 4D) and PAQR7 (Figure 4E). These results demonstrate that PAQR7, PGRMC1, and nPRs can be used as an HCC specific biomarker, while CCM1/3 and PAQR8 can be used as a biomarker signature to differentiate undifferentiated pleomorphic sarcoma (UPS) and hepatic angiosarcoma (HAS) from other hepatic cancer subtypes.

#### Altered Expression Levels of CCM3 and PAQR7 in Biphenotypic Liver Cancer Subtype 

Furthermore, we compared CCM3 and PAQR7 protein expression between HCC and cHCC-CCA (Figure 5 and Appendix A), since cHCC-CCA is known to be a biphenotypic liver cancer subtype, presenting hepatocytic and biliary differentiation [53]. IF methods revealed cHCC-CCA tissues exhibited the lowest abundance of both proteins compared to NORM and HCC tissues (Figure 5(A-1),A-2)). While examining CCM3 and PAQR7 expression, we observed NORM tissue samples contained more co-localized “clumped” patterns for CCM3 and PAQR7 (Figure 5B—60× imaging) compared to both HCC and cHCC-CCA tissues, suggesting down regulation of key members of the CmPn network during liver tumorigenesis. Our data also indicate the loss of specific cellular co-localization patterns of both CCM3 and PAQR7 can be used as a pathological biomarker signature for diagnosing cHCC-CCAs and HCCs. Overall, these results demonstrate the potential prognostic application of CmPn members’ expression data as biomarkers for distinguishing between primary subtypes of liver cancer and evaluating tumor recurrence in patients.

### 3.7. Copy Number Variations of CmPn Players in HCC Tissues Confirms Crosstalk among the CmPn Signaling Network during Liver Cancer Tumorigenesis

To evaluate the relationship between key CmPn players within the CmPn signaling network, we utilized publicly available microarray data from 14 HCC patients to generate an Oncogrid (Figure 6A). Each column contains copy number variation (CNV) information, mutation type for each CmPn gene, as well as clinical information. Our analysis identified key CmPn mutations seen in HCC patients, but also solidified the existence of crosstalk interactions among key members of the CmPn network during liver tumorigenesis. We observed not only missense (GOF or LOF), frameshift, and nonsense (LOF) CmPn gene mutations, but also observed compound mutations (one GOF and one LOF) that co-exist among CmPn genes (Figure 6A). Both *CCM1* and *PAQR6* were the two members impacted with gain of function (GOF) alterations, while *nPRs* was the only member frequently impacted with a loss of function (LOF) alteration (Figure 6A). Next, we wanted to assess clinical impacts of altered expression levels of key CmPn members. These findings suggest that CNVs and mutations in the CmPn signaling network may contribute to the development and progression of HCC, further emphasizing the importance of this network in liver tumorigenesis.

### 3.8. Differential Expression Patterns of mPR and CCM Genes across Clinical Tumors and Their Associated Survival Effects

Utilizing publicly available microarray data from HCC patients (not enough data available for CCAs), we incorporated gene expression and clinical data simultaneously to generate Kaplan–Meier (KM) survival curves to understand the impact of the CmPn network on overall survival (OS) in liver cancer patients (Figure 6B). In each panel, the left KM curve was generated using KMplotter [43], while the right KM curve in each panel was generated using TCGA data to validate our KMplotter analysis. Our analysis revealed increased expression of *CCM3* (Figure 6(B-1)) and *PAQR5/7/9* (Figure 6(B-3)), while decreased expression of *PGRMC1/2* (Figure 6(B-2)) demonstrated significantly worse prognostic outcomes in HCC patients. Of these six key CmPn players, we were able to validate all survival curves, except *PAQR5,* utilizing TCGA data. These results further substantiate the involvement of the CmPn signaling network in liver cancer tumorigenesis and further solidify their potential prognostic application in HCCs. In sum, by integrating with survival curves from TCGA database, these findings provide additional support for the potential use of CmPn genes as prognostic biomarkers for HCC patients.

## 4. Discussion

Recent studies have highlighted the crucial role of the CmPn signaling network in the development of multiple types of cancer [16,17,22,23,26,27,28]. It has been demonstrated that the mPRs and nPRs play equally important roles in PRG signaling and that organs that metabolize PRG are susceptible to changes in CmPn members’ expression during tumorigenesis [12]. The novel CmP signaling network (absence of nPRs), in African American women-derived triple negative breast cancer (TNBC) cells [17], aligned with observations of altered CSC and mPRs expression changes in nPRs(+) breast cancer cells [18]. Studies using Caucasian American women-derived TNBC cells showed that all three breast cancer subtypes exhibited alterations to key tumorigenesis pathways, including cytokine-mediated responses, Wnt, integrin, gonadotropin-releasing hormone, and angiogenesis signaling pathways, under mPR-specific PRG actions (PRG actions through mPRs only) [16]. In relation to liver tumorigenesis, a previous study found the largest differential expression of CmPn members in liver tumor tissues, followed by endometrial and breast tumor tissues [26]. In this study, using RNAseq data and IF approaches, we further validated the importance of the CmPn signaling network in liver cancer tumorigenesis by examining the expression profiling of key CmPn members at both the transcriptional and translational levels across multiple liver cancer subtypes.

Our findings revealed altered expression of several CmPn members (including CCM1/3 and PAQR7/8) between normal and tumor tissues and among different liver cancer subtypes, which was also observed for PGRMC1 (Figure 4D). Additionally, we found altered expression of nPRs (Appendix A) in all liver cancer subtypes, compared to normal healthy liver tissues, and between HCC and all other subtypes (Appendix A). Interestingly, we identified AFP as a potential biomarker specific to CCA tissues, with decreased expression observed at both the transcriptional (Figure 2(D-1)) and translational levels (Appendix A), as opposed to the well-documented trends observed in HCC.

We utilized systems biology approaches to compare the alterations of key tumorigenesis pathways in CCAs and HCCs with normal healthy tissues, revealing similarities to those observed in our breast cancer omics analysis [16,17,19,20,25]. Specifically, we observed dysregulation of common cancer signaling pathways that influence liver tumorigenesis, including sterol/cholesterol homeostasis [54], bile secretion [55,56], ABC transporters [57], PPAR signaling [58], and frizzled binding [59] (Figure 2C,D). Moreover, we found that CCAs and HCCs have opposite regulation in these signaling cascades, with cholesterol/triglyceride/lipid homeostasis, cholesterol transport, PPAR signaling, bile secretion, and ABC transporters down regulated in CCAs compared to HCCs, while frizzled binding, MAPKKK signaling, and protein kinase binding were upregulated in CCAs (Figure 2E). These results highlight the importance of evaluating altered gene expression of key CmPn members involved in these signaling cascades [16,17,19,20,25].

### 4.1. Potential Prognostic Biomarkers for Hepatic Cancers

Biomarkers play a crucial role in the prognosis and management of liver cancer, aiding in early detection and predicting patient response to therapy and the course of the disease [60,61]. With 5-year survival rates for liver cancer patients ranging from 5 to 30%, early detection is critical for improving patient outcomes, especially for those with HCC and CCA, which have survival rates of 15% and 5–15%, respectively [4]. Timely treatment for HCC has been shown to result in longer recurrence-free periods and better overall survival rates [5]. However, HCC has been associated with resistance to chemotherapy, making it difficult to treat and resulting in poor prognosis [6]. Therefore, the identification of sensitive and specific biomarkers can aid in the early detection and management of liver cancer, ultimately improving patient outcomes.

The current lack of reliable biomarkers for the diagnosis and prognosis of hepatic cancers underscores the need for identifying new biomarkers for various subtypes to improve early detection and treatment options for patients [34,61,62,63,64]. While AFP is the gold standard, it has limitations in detecting different grades and stages of HCC and accurately differentiating between primary liver cancer types [34,65,66]. Additionally, rarer liver cancer subtypes have even fewer available options for early detection. Therefore, our observation of differential expression of multiple members of the CmPn network in both HCC and CCA clinical tumor samples, at both the transcriptional and translational levels, highlights the potential for these members to serve as novel biomarkers for liver cancer. These findings could lead to the development of more sensitive and specific biomarkers to aid in early diagnosis, response to therapy, and prognosis for patients with liver cancer.

Utilizing RNA-seq and IF approaches, we observed that *CCM1* and *nPRs* were seen to be down regulated, at both the transcriptional and translational levels, in HCC and CCA tissues when compared to normal tissues, indicating that these CmPn members could be useful diagnostic biomarkers, but did not support its potential use as prognostic biomarkers in differentiating between cancer subtypes. Up-regulated levels of *PAQR7* in HCC tissues only, at both the transcriptional (Figure 2(C-1)) and translational levels (Figure 4E), as well as our observation that increased expression of *PAQR7* is associated with decreased survival in HCC patients (Figure 6(B-3)), support its use as a potential prognostic biomarker to differentiate between HCCs and CCAs. Previous research has associated decreased expression of *PGRMC1* with advanced stages and poorer prognosis, and therefore has been proposed as a potential prognostic biomarker for HCC and various other cancers [32,67,68]. We observed down regulation of PGRMC1 in HCCs at the translational level (Figure 4D) but observed that mRNA levels of PGRMC1 are actually slightly increased compared to normal healthy tissues (Figure 2(A-1),(C-1)). Interestingly, we observed more dramatic down regulation of *PGRMC1,* occurring at both the transcriptional (Figure 2(A-2),(D-1)) and translational levels (Figure 4D), in CCAs, further suggesting its potential use as a prognostic biomarker for CCAs. These findings are extremely significant since CCAs have less available validated biomarkers for early detection [66]. Interestingly, in addition to analyzing expression levels of key CmPn members, we assessed AFP expression levels in CCA tissues and saw synchronous down regulation at both the transcriptional (Figure 2(A-1),(D-1)) and translational levels (Appendix A), which corroborates with previous findings that *AFP* levels are typically lower in CCAs than normal tissues, contrary to increased expression observed in HCCs [34]. These results are significant, particularly for CCAs, which have fewer validated biomarkers available for early detection.

### 4.2. Pathways Functional Enrichment Analysis Using DEGs for HCCs and CCAs Demonstrate Perturbation in Key Tumorigenic Signaling Cascades

Pathways functional enrichment results obtained from significantly altered genes in HCC clinical samples demonstrated perturbation to key cancer signaling pathways including cell cycle, DNA replication, spliceosomes/lysosomes, pancreatic cancer signaling pathways, and pancreatic secretion pathways (Figure 2(C-2)). Interestingly, we also observed dysregulation in hormone signaling pathways including oocyte meiosis and PRG-mediated oocyte maturation (Figure 2(C-2)), which were also altered pathways observed in all of our breast cancer studies [16,17,23,27,28]. Functional enrichment results obtained from significantly altered genes in CCA clinical samples demonstrated perturbation to key cancer signaling pathways including cell population proliferation/differentiation, cell junction assembly, Wnt signaling, PI3K-Akt, and Notch signaling (Figure 2(D-2)), which were also observed to be dysregulated in all of our breast cancer studies [16,17,23,27,28]. We also observed newly dysregulated pathways, not observed in our breast cancer studies, including disruption of ECM–receptor interactions, complement/coagulation cascades, PPAR pathways, bile secretion, peroxisomes, sterol/cholesterol homeostasis, and ABC transporters (Figure 2(D-2)), which are known to influence liver tumorigenesis. Interestingly, when we repeated our analysis to compare CCAs with HCCs, we noticed opposite regulation between the two common liver cancer subtypes including down regulation of cholesterol/triglyceride/lipid homeostasis, cholesterol transport, PPAR signaling, bile secretion, as well as ABC transporters in CCAs compared to HCCs (Figure 2(E-2)). Similarly, frizzled binding, MAPKKK signaling, and protein kinase binding were observed to be up regulated in CCAs compared to HCCs (Figure 2(E-2)), solidifying the importance of these common signaling cascades in distinguishing HCCs from CCAs. Overall, the findings provide important insights into the molecular mechanisms underlying liver cancer development and progression and highlight the need for continued research in this area to improve patient outcomes.

### 4.3. Pathways Functional Enrichment Analysis Using DEGs in the Immune Landscape of Hepatic Cancers

Given the importance of tumor microenvironments for both tumorigenesis and immunogenicity in hepatic cancers [50], one of our main tasks was also to assess differential expression of CmPn network genes, along with *AFP*, among immune subtypes for both HCCs and CCAs.

#### 4.3.1. Wound Healing (C1) Subtype Compared to IFN-γ Dominant (C2) Subtype

C1 (wound healing) immune subtypes are characterized by elevated expression of angiogenic genes, while C2 (IFN-γ dominant) subtypes have high M1/M2 macrophage polarization [51]. Cirrhosis is a form of recurrent wound healing and, when seen in liver cancer patients, can decrease life expectancy [69]. An M2 to M1 phenotypic shift occurs during HCC tumor progression, which promotes cancer cell proliferation [70]. Intriguingly, we noted during our comparisons that there were very few significant up-regulated pathways in HCCs, while CCAs demonstrated significance for both up- and down-regulated pathways (Appendix A). When comparing the two immune subtypes, we observed shared down-regulated pathways in C1 subtype, for both HCCs and CCAs, including chemokine/cytokine receptor activity, MHC protein complex binding, Th1/2/17 cell differentiation, and CAMs pathways (Appendix A). Interestingly, down regulation of TNF-activated receptor activity and NF-kappa B signaling were only observed in CCAs (Appendix A), both of which are associated with aggressive CCA development and poor prognosis [71,72].

#### 4.3.2. Wound Healing (C1) Subtype Compared to Inflammatory (C3) Subtype

We next compared C1 immune subtype with the C3 (inflammatory) immune subtype, characterized by elevated Th1 and Th17 genes and low to moderate tumor cell proliferation [51]. High levels of Th1 and Th17 are associated with HCC invasiveness and can be indicative of increased tumor progression [73]. Interestingly, among the top altered pathways in C1 subtype, we did not observe any shared up-regulated pathways between HCCs and CCAs (Appendix A). Interestingly, shared down-regulated pathways in C1 subtype, for both HCCs and CCAs, included peroxisomes, glycine/serine/threonine metabolism, and PPAR signaling pathways (Appendix A), of which PPARγ has been demonstrated to display inhibitory effects on HCC metastasis [74,75].

#### 4.3.3. Wound Healing (C1) Subtype Compared to Lymphocyte Depleted (C4) Subtype

Similarly, we compared C1 subtype to the C4 (lymphocyte depleted) subtype, which is classified as having suppressed Th1 levels and a high M2 response [51]. An elevated M2 response in HCC has been associated with tumor resistance to sorafenib, which is currently utilized as a standard treatment for liver cancers [76]. We observed shared up-regulated pathways in C1 subtype, for both HCCs and CCAs, including AGE-RAGE signaling, ECM organization, and proteoglycans in cancer (Appendix A). Interestingly, previous studies have shown that AGE-RAGE signaling is up regulated in liver fibrosis, while RAGE silencing reduces liver tumor growth [77]. Shared down-regulated pathways in C1 subtype, for both HCCs and CCAs, included fatty acid degradation and lipid/lipoprotein metabolism (Appendix A). To our surprise, Wnt signaling was observed to be up regulated only in HCCs (Appendix A), which was also a main altered pathway observed in our previous breast cancer studies investigating effects of altered CmPn expression [16,17,23,27,28].

#### 4.3.4. IFN-γ Dominant (C2) Subtype Compared to Inflammatory (C3) Subtype 

In our analysis, we observed several shared up-regulated pathways in C2 subtype, for both HCCs and CCAs, including cytokine-mediated signaling, CAMs, and Th17 cell differentiation pathways (Appendix A). Previous studies have demonstrated that the production of IL-17 by Th17 cells induces liver inflammation by stimulating multiple types of liver nonparenchymal cells that secrete cytokines and chemokines [78]. Interestingly, shared down-regulated pathways in C2 subtype, for both HCCs and CCAs, included Rap1 signaling, cadherin binding, and adherens junction pathways (Appendix A). Previous studies have shown that Rap1 suppresses tumorigenesis and is up regulated in HBV-related HCC pathogenesis [79]. Intriguingly, down regulation of FoxO signaling pathways in C2 subtype was only observed for HCCs (Appendix A), which have been associated with poor prognosis in liver cancer patients, as they develop resistance to chemotherapy treatment [80].

#### 4.3.5. IFN-γ Dominant (C2) Subtype Compared to Lymphocyte Depleted (C4) Subtype

Based on our analysis, we observed shared up-regulated pathways in C2 subtype, for both HCCs and CCAs, including cytokine/chemokine receptor activity, CCR chemokine receptor binding, and hematopoietic cell lineage pathways (Appendix A). Interestingly, recent studies have shown that overexpression of certain CCRs in HCC tissues correlates with vascular invasion [81]. Shared down-regulated pathways in C2 subtype, for both HCCs and CCAs, included fatty acid degradation, valine/leucine/isoleucine degradation, and pyruvate metabolism (Appendix A). Interestingly, down regulation of SIRT1 negative regulation of rRNA expression was only observed for C2 subtype in HCCs (Appendix A), which has been found to be an independent prognostic factor for HCCs, as SIRT1 promotes liver tumor cell survival [82].

#### 4.3.6. Inflammatory (C3) Subtype Compared to Lymphocyte Depleted (C4) Subtype

For our final comparison, we observed shared up-regulated pathways in C3 subtype, for both HCCs and CCAs, including integrin binding, general pathways in cancer, as well as AGE-RAGE signaling pathways (Appendix A). We also observed shared down-regulated pathways in C3 subtype, for both HCCs and CCAs, including proteasomes, oxidative phosphorylation, and translation initiation factor activity pathways (Appendix A). Finally, similar to our previous study analyzing altered CmP expression levels and resulting effects on signaling pathways in African American women-derived TNBCs [17], we again observed altered PI3K-Akt, MAPK, and angiogenesis pathways for C3 subtype, but only in HCCs (Appendix A).

These findings suggest the dysregulated pathways observed in both immune subtypes, as well as the differences in altered pathways between the subtypes, highlight the importance of understanding the immune landscape in hepatic cancers and the potential role of CmPn members as prognostic biomarkers.

### 4.4. Further Validation of the CmPn Signaling Network in Liver Cancer

As previously mentioned, differential expression of key CmPn members has been shown to be involved in the progression of multiple types of cancers, as well as leading to worse prognostic outcomes [29,30,32,67]. The role of mPRs in tumorigenesis has been increasingly studied, more recently in liver cancer given that the liver is a primary target and metabolic organ of PRG [12]. Previous research has demonstrated that *PAQR8* and *PAQR9* are independent prognostic factors for HCC [83], which was supported in our RNA-seq analysis, as well as our generated KM survival curves for *PAQR9,* which demonstrated increased expression resulted in decreased overall survival (Figure 6(B-3)). Finally, CNV analysis of HCC patients with confirmed CmPn genomic (either GOF or LOF) mutations could lead to perturbation of the CmPn signaling network in hepatic cancers. Together, these findings provide further insight into the complex interactions among key members of the CmPn network in liver cancer and may have implications for the development of novel therapies for this disease.

## 5. Conclusions

The findings in this study are significant because hepatic cancers are a major cause of cancer-related deaths worldwide, and early diagnosis and treatment can greatly improve patient outcomes. The identification of potential prognostic biomarkers and the discovery of disrupted signaling pathways and altered CmPn members in different subtypes of hepatic cancers can lead to the development of targeted therapies that could improve patient outcomes and potentially lead to better overall survival rates. Additionally, these results provide insights into the molecular mechanisms of liver cancer tumorigenesis, which may lead to a better understanding of the disease and new strategies for its prevention and treatment.

## Figures and Tables

**Figure 1 diagnostics-13-01012-f001:**
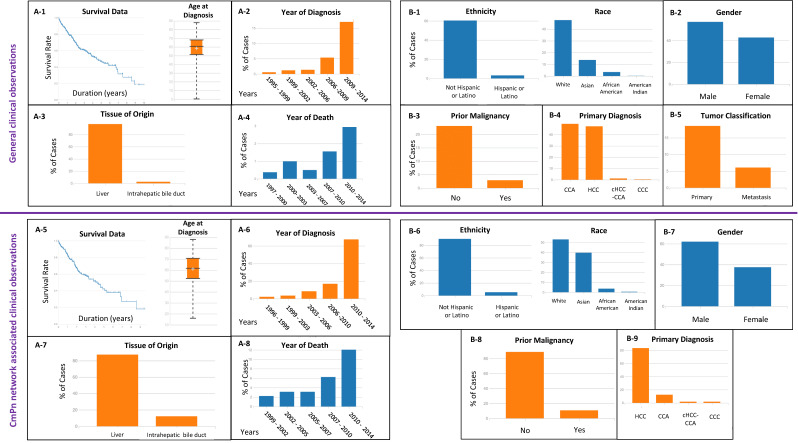
Clinical profiling for hepatic cancers utilizing NCI-GDC data. Utilizing the Genomic Data Commons (GDC) data portal from the National Cancer Institute (NCI), we assessed all available clinical data for patients diagnosed with hepatic cancers without any gene filters (general clinical observations, panels (**A1–4**,**B1–5**)) or filtered to only analyze patient samples with differential expression for any of the key CmPn players (CmPn network associated clinical observations, panels (**A5–8**,**B6–9**)). (**A-1**) Survival data (n = 420, left panel) and age at diagnosis (n = 829, right panel). (**A-2**) Year of diagnosis (automatically divided into 5 categories; n = 423). (**A-3**) Tissue of origin (n = 1606). (**A-4**) Year of death (automatically divided into 5 categories; n = 106). (**A-5**) Survival data (n = 219, left panel) and age at diagnosis (n = 220, right panel). (**A-6**) Year of diagnosis (n = 221). (**A-7**) Tissue of origin analysis (n = 223). (**A-8**) Year of death analysis (n = 60 total). *Socioeconomic status (SES) and diagnostic profiling for hepatic cancers utilizing clinical data from the NCI-GDC data portal.* (**B-1**) Ethnic (n = 1022, left panel) and primary racial categories (n = 1047, right panel). (**B-2**) Gender analysis (n = 1606). (**B-3**) Prior malignancy status (for available data; n = 419). (**B-4**) Primary diagnosis (liver vs. intrahepatic bile ducts; n = 1582). (**B-5**) Tumor classification (n = 397 total). (**B-6**) Ethnic (n = 214, left panel) and primary racial category analysis (n = 217, right panel). (**B-7**) Gender analysis (n = 223). (**B-8**) Prior malignancy analysis (n = 223). (**B-9**) Primary diagnosis analysis (n = 223 total). For all graphs, the comparisons were performed automatically by the NCI-GDC software, for all publicly available clinical data, utilizing 14 projects (for general clinical observations) or 2 projects (for CmPn network associated clinical observations) for patients diagnosed with hepatic cancers (see Appendix A). Abbreviations: cholangiocarcinoma (CCA), hepatocellular carcinoma (HCC), combined HCC+CCA (cHCC-CCA), clear cell carcinoma (CCC).

**Figure 2 diagnostics-13-01012-f002:**
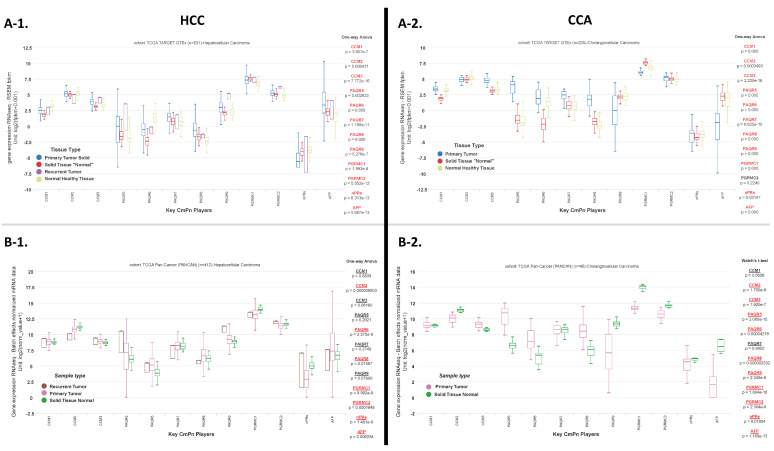
RNAseq expression profiling for key CmPn players utilizing multiple TCGA databases for hepatic cancers. We performed expression profiling for key *CSC*, *mPR*, and *nPR* players, along with established liver cancer biomarker *AFP*, using two types of ‘normal’ tissues utilizing the TCGA-TARGET-GTEX and TCGA-PANCAN databases: solid tissue “normal” which are taken from “normal” tissue, near the tumor site, and normal healthy tissue from individuals without cancer. (**A1–2**) Differentially expressed genes (DEGs) of key CmPn players (along with *AFP*) for HCCs (n = 531) and CCAs (n = 205), based on tissue type using the TCGA-TARGET-GTEX database. (**B1–2**) Differential expression of key CmPn players (along with *AFP*) for HCCs (n = 412) and CCAs (n = 45), based on tissue type using the TCGA-PANCAN database. (**C-1**) Significant DEGs were profiled in a pair-wise fashion between normal healthy liver tissue (n = 110) and HCC primary tumors (n = 369). (**C-2**) Pathway functional enrichment comparisons were performed using Enrichr assessing biological processes (upper panel), molecular functions (middle panel), and KEGG signaling pathways (lower panel). (**D-1**) Significant DEGs were profiled in a pair-wise fashion between normal healthy liver tissue (n = 110) and CCA primary tumors (n = 36). (**D-2**) Pathway functional enrichment comparisons were performed assessing biological processes (upper panel), molecular functions (middle panel), and KEGG signaling pathways (lower panel). (**E-1**) Significant DEGs were profiled from a separate TCGA database, with increased sample sizes, between HCC (n = 438) and CCA (n = 45) tumors. (**E-2**) Pathway functional enrichment comparisons were compiled using biological processes (upper panel), molecular functions (middle panel), and KEGG signaling pathway data (lower panel). For all boxplots, X-axis details genes profiled, while Y-axis details normalized RNAseq expression data; significantly altered genes were identified using one-way ANOVA analysis and are color-coded red. For Panels C-E, red colored genes were up regulated while blue colored genes were down regulated; for pathways analysis, the top 15 pathways are provided and color coded (red—up/blue—down) and pathways with *p*-values < 0.05 and FDR < 0.1 are bolded. All graphs were produced using the Xena platform.

**Figure 3 diagnostics-13-01012-f003:**
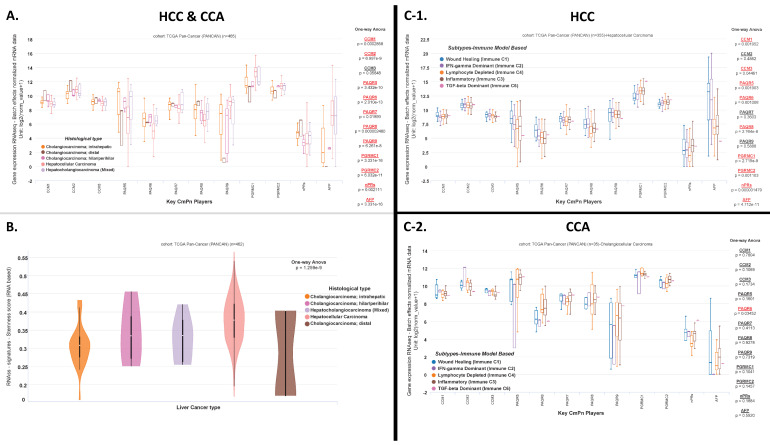
RNAseq expression profiling for key CmPn players for HCC and CCA histological/immunological subtypes integrating SES and follow-up clinical data. We investigated key *CSC*, *mPR*, and *nPR* players expression analysis, along with *AFP*, between histological and immunological subtypes (panels **A**,**C1**,**C2**) and assessed stemness scores (panel **B**) between major HCC and CCA subtypes. Additionally, we performed expression profiling using demographic and follow-up data to assess differential expression across major races and evaluate impact on tumor recurrence (panels (**D–I**)). (**A**) Significant DEG profiling between the three major subtypes of CCA compared to HCC and cHCC-CCA clinical tumors (n = 465). (**B**) Stem cell-associated molecular scores between the three major subtypes of CCAs compared to HCC and cHCC-CCA clinical tumors (n = 462). (**C-1**) Significant DEG profiling between standard immune subtype classifications for HCCs (n = 355). (**C-2**) Significant DEG profiling between standard immune subtypes for CCAs (n = 35). (**D**) Significant DEG profiling among the 4 most prevalent races diagnosed with HCC (n = 399). (**E**) Significant DEG profiling for patients diagnosed with HCC and with documented family cancer history (n = 358). (**F**) Significant DEG profiling for HCC patients presenting with either micro/macro or no vascular invasion (n = 412). (**G**) Significant DEG profiling for HCC patients presenting with a new primary tumor, locoregional [cancer cells at the same site as the original (primary) tumor], intrahepatic (recurrence in liver after hepatic resection), or extrahepatic recurrence (most commonly in the lungs, lymph nodes, and bones, n = 174). (**H**) Significant DEG profiling for HCC patients among the 5 major locations for tumor recurrence (n = 150). (**I**) Significant DEG profiling for HCC patients based on vital status (n = 412). For all boxplots (panels **A**,**C**,**D**–**I**), X-axis details genes profiled, while Y-axis details Log2 normalized RNAseq expression data. For Panel B, X-axis details HCC and CCA subtypes profiled, while Y-axis details RNAss-stemness signature scores (RNA-based). All graphs were produced using the Xena platform.

**Figure 4 diagnostics-13-01012-f004:**
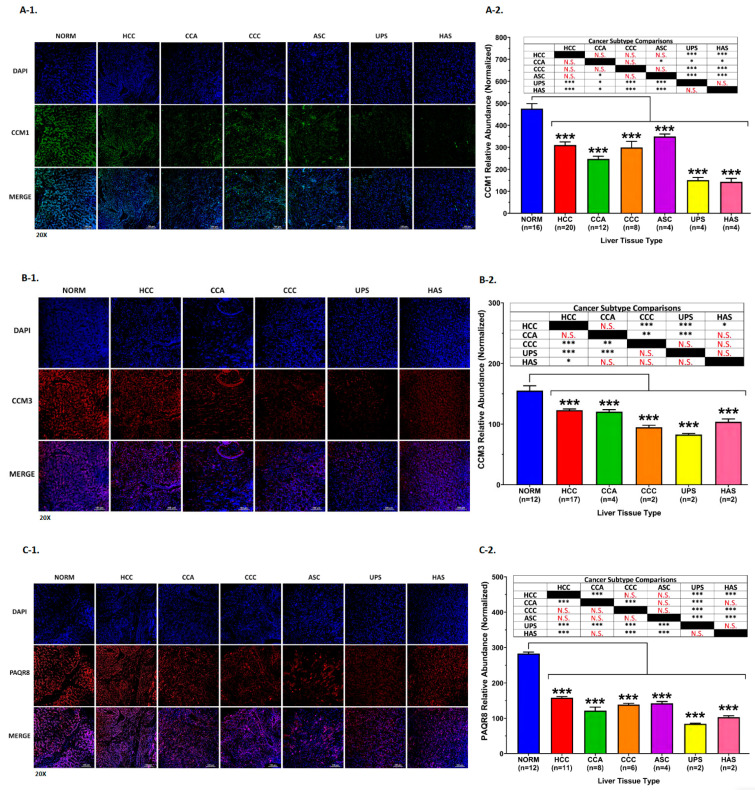
Significantly altered CmPn protein expression among hepatic cancer subtypes. Comparative CmPn protein expression patterns were measured with immunofluorescence-labeled antibodies, normalized against nuclear staining (DAPI), and quantified using Nikon Elements Analysis software. (**A-1**) Representative IF images of CCM1 protein expression among different liver cancer types and NORM tissues. (**A-2**) Normalized quantification for all images. (**B-1**) Representative IF images of CCM3 protein expression among different liver cancer types and NORM tissues. (**B-2**) Normalized quantification for all images. (**C-1**) Representative IF images of PAQR8 protein expression among different liver cancer types and NORM tissues. (**C-2**) Normalized quantification for all images. (**D-1**) Representative IF images of PGRMC1 protein expression among different liver cancer types and NORM tissues. (**D-2**) Normalized quantification for all images. **(E-1**) Representative IF images of PAQR7 protein expression among different liver cancer types and NORM tissues. (**E-2**) Normalized quantification for all images. For all graphs, asterisks above each cancer type indicate significance of differential expression between specific cancer type and NORM tissues. Cancer subtype comparisons chart above each graph summarizes the relationships within cancer subtypes. Sample sizes provided beneath liver tissue type. In all bar plots and charts, ***, **, and * indicates *p* ≤ 0.001, ≤0.01, and ≤0.05, respectively, N.S. means non-significant, using one-way ANOVA.

**Figure 5 diagnostics-13-01012-f005:**
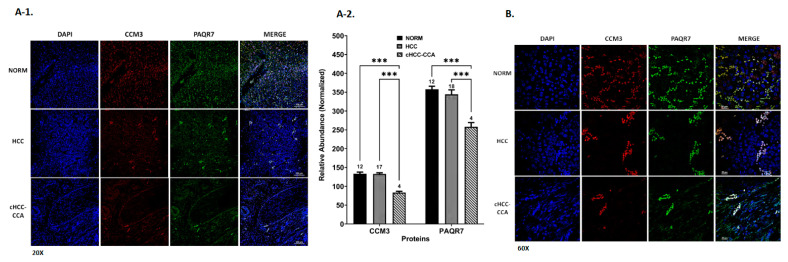
Relative abundance of CCM3 and PAQR7 between HCC and cHCC-CCA subtype. CCM3 and PAQR7 protein expression patterns were measured with immunofluorescence-labeled antibodies, normalized against nuclear staining (DAPI), and quantified using Nikon Elements Analysis software. IF imaging was conducted at 20× (left panels) and 60× magnification (right panels) on NORM, HCC, and cHCC-CCA tissues. (**A-1**) Representative IF images of CCM3 and PAQR7 protein expression for NORM, HCC, and cHCC-CCA tissues captured at 20x magnification. (**A-2**) Normalized quantification of CCM3 and PAQR7 for NORM, HCC, and cHCC-CCA tissues. (**B**) Representative 60× IF imaging of areas from panel A to further examine clustering patterns of key CmPn proteins in liver tissues. Sample sizes provided above bar graphs. In all bar plots, *** indicates *p* ≤ 0.001, using two-way ANOVA.

**Figure 6 diagnostics-13-01012-f006:**
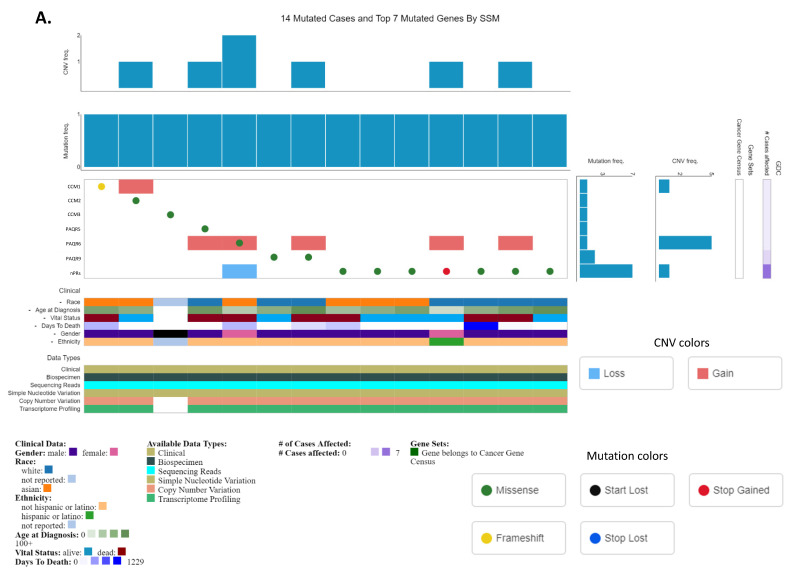
Oncogrid with copy number variations (CNV) and prognostic effects for key CmPn players utilizing microarray data for HCC patients. Utilizing the Genomic Data Commons (GDC) data portal from the National Cancer Institute (NCI), we assessed all available clinical and CNV data for patients diagnosed with hepatic cancers with differential expression for any of the key CmPn players. Additionally, publicly available microarray data from HCC patients were analyzed using either KMplotter or Xena browser (TCGA database) to integrate gene expression and clinical data simultaneously to generate Kaplan–Meier (KM) survival curves. (**A**) CNV analysis for HCC patients with CmPn mutations. The clinical data legend is provided beneath the Oncogrid, as well as coloring scheme for mutation outcomes and CNV impacts. (**B-1**) Prognostic effects for *CCM3* in both KMplotter (left panel) and TCGA databases (right panel). (**B-2**) Prognostic effects for *PGRMC1/2* in both KMplotter (left panels) and TCGA databases (right panels). (**B-3**) Prognostic effects for *PAQR5/7/9* in KMplotter (left panels) and TCGA databases (right panels). For Panels B1-3, log-rank *p*-values were automatically calculated and displayed for KMplotter analysis as well as hazard ratios (and 95% confidence intervals). Alternatively, *p*-values and log-rank test statistics were automatically calculated and displayed for TCGA analysis. Red line demonstrates high gene expression (all panels), while black line (TCGA analysis) or blue line (KMplotter analysis) demonstrates low gene expression.

## Data Availability

Readers can access the data supporting the conclusions of the study through Appendix A and some omics data are in process to deposit into NIH genomic or proteomic databases repertoire and can be acquired by contacting the corresponding author.

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
