# Peer review of "Key Members of the CmPn as Biomarkers Distinguish Histological and Immune Subtypes of Hepatic Cancers"

_diagnostics, 2023, doi:10.3390/diagnostics13061012_

Round 1

Reviewer 1 Report

this article discusses a hot topic in the field of hepatic tumor but much work and double blind randomized studies are needed to achieve these goals.

regarding figures, add annotations, scale bar, type of stain or dye. also, use high quality figures with good brightness.

Reviewer 2 Report

The authors have conducted an interesting study and could efficiently use the bioinformatics databases for the analysis of CmPn members in liver cancer. There are a vast amount of precious data, but it is difficult for the readers to follow all data. I suggest making the manuscript concise and focusing on the study's main subject.  

Also, there are minor revisions: 

1- In 70-73 authors have reported two contrasting alterations. Is it clear if upregulation or downregulation of the members of the CmPn pathway promotes tumorigenesis? 

2- In line 82, the authors mentioned biomarkers for "provide non-invasive prognostic methods to distinguish between liver cancer 82 subtypes". 

We usually use prognostic biomarkers for obtaining information about the outcome of therapy. Do we use prognostic biomarkers for distinguishing cancer subtypes? 

3- In line 135: Actually 90 minutes is not a fast protocol for blocking. Please recheck the manufacturer's protocol

4- Line 136: The dilutions of the primary and secondary antibodies should be mentioned. Also, I could not find the list of used antibodies in supplementary files (Table 3). 

5- Line 144: "Tissues were left to rest O/N at 4℃ in the dark to allow efficient staining". What does "O/N" mean here? 
